# Green Biogenic of Silver Nanoparticles Using Polyphenolic Extract of Olive Leaf Wastes with Focus on Their Anticancer and Antimicrobial Activities

**DOI:** 10.3390/plants12061410

**Published:** 2023-03-22

**Authors:** Bassam F. Alowaiesh, Haifa Abdulaziz Sakit Alhaithloul, Ahmed M. Saad, Abdallah A. Hassanin

**Affiliations:** 1Olive Research Center, Jouf University, Sakaka 72341, Saudi Arabia; 2Biology Department, College of Science, Jouf University, Sakaka 72341, Saudi Arabia; 3Biochemistry Department, Faculty of Agriculture, Zagazig University, Zagazig 44511, Egypt; 4Genetics Department, Faculty of Agriculture, Zagazig University, Zagazig 44511, Egypt

**Keywords:** olive leaves, wastes, extraction, nanoparticles, cancer, MDR microorganisms, medical applications

## Abstract

Agro-industrial wastes are rich in polyphenols and other bioactive compounds, and valorizing these wastes is a crucial worldwide concern for saving health and the environment. In this work, olive leaf waste was valorized by silver nitrate to produce silver nanoparticles (OLAgNPs), which exhibited various biological, antioxidant, anticancer activities against three cancer cell lines, and antimicrobial activity against multi-drug resistant (MDR) bacteria and fungi. The obtained OLAgNPs were spherical, with an average size of 28 nm, negatively charged at −21 mV, and surrounded by various active groups more than the parent extract based on FTIR spectra. The total phenolic and total flavonoid contents significantly increased in OLAgNPs by 42 and 50% over the olive leaf waste extract (OLWE); consequently, the antioxidant activity of OLAgNPs increased by 12% over OLWE, recording an SC_50_ of OLAgNPs of 5 µg/mL compared to 30 µg/mL in the extract. The phenolic compound profile detected by HPLC showed that gallic acid, chlorogenic acid, rutin, naringenin, catechin, and propyl gallate were the main compounds in the HPLC profile of OLAgNPs and OLWE; the content of these compounds was higher in OLAgNPs than OLWE by 16-fold. The higher phenolic compounds in OLAgNPs are attributable to the significant increase in biological activities of OLAgNPs than that of OLWE. OLAgNPs successfully inhibited the proliferation of three cancer cell lines, MCF-7, HeLa, and HT-29, by 79–82% compared to 55–67% in OLWE and 75–79% in doxorubicin (DOX). The preliminary worldwide problem is multi-drug resistant microorganisms (MDR) because of the random use of antibiotics. Therefore, in this study, we may find the solution in OLAgNPs with concentrations of 2.5–20 µg/mL, which significantly inhibited the growth of six MDR bacteria *L. monocytogenes, B. cereus, S. aureus, Y. enterocolitica, C. jejuni,* and *E. coli* with inhibition zone diameters of 25–37 mm and six pathogenic fungi in the range of 26–35 mm compared to antibiotics. OLAgNPs in this study may be applied safely in new medicine to mitigate free radicals, cancer, and MDR pathogens.

## 1. Introduction

Nanotechnology is a new science branch, achieving other successes in life sciences, particularly in medicine and agriculture [1,2]. Silver nanoparticles (AgNPs) are widely utilized in several applications such as home disinfectants, medical devices, water purificators, the food industry, health care, and other commercial products. Caution must be taken regarding their potential toxicity, safety, and risks [3]. The size of AgNPs is tiny, ranging between 10 and 100 nm. The nanoparticles are produced through different physical or chemical methods, but these routes consume high energy and have toxic effects, so an eco-friendly biological path has emerged [4]. Other studies have synthesized nanoparticles using bacteria, algae, and plants [5,6,7]. Therefore, the environmentally friendly synthesis of silver nanoparticles confirms their risk-free use in various biological applications [8] because they exhibit anticancer, antimicrobial, and antioxidant activities [9].

The current study used olive leaf waste extract to synthesize AgNPs because it contains many bioactive components and groups involved in biotransformation [10]. The olive leaves contain caffeic acid, luteolin 7-O-glucoside, oleuropein, verbascoside, rutin, luteolin 4′-O-glucoside, apigenin 7-O-glucoside, and hydroxytyrosol, which have been demonstrated to suppress the development of *S. aureus*, *E. coli*, and *K. pneumoniae* when isolated from olive leaves. Oleuropein, a key ingredient obtained from olive leaves, has been discovered to inhibit *B. cereus* sporulation. Hydroxytyrosol, a metabolite of oleuropein, has recently been shown to be effective against *S. typhimurium* and *S. aureus*, which are clinically harmful to humans. Tyrosol was determined to be an effective inhibitor of the microorganisms *L. monocytogenes* and *A. flavus*. Rutin is effective against *P. aeruginosa* bacteria, *Candida albicans*, and *Candida neoformans*. Several studies have revealed that the phenolic components of olive leaf extract, particularly oleuropein and hydroxytyrosol, have powerful antioxidant and anticancer properties [11,12,13].

Recent years have witnessed the emergence of microbial infections with significant resistance to even the strongest medications. This has led to a considerable burden on the health system and increased worldwide mortality and morbidity rates. The critical antibiotic discovery pipeline and the widespread use of antibiotics have sounded the alarm, needing rapid action to counteract this impending threat. Diverse strategies for creating novel medicines such as countering antibiotic resistance are gaining prominence and are revitalizing the existing arsenal. Numerous investigations have identified substances that lower the permeability barrier, resistance-encoding enzymes, and the expression of efflux pumps. Compounds that target the stability, transfer, and dispersion of the transposable genetic compounds associated with gene resistance are prospective antibacterial agents. Diverse natural sources and synthetic chemicals have been used in the quest for such molecules [14].

The properties of the AgNPs produced by plants are unique and safer than those chemically produced, which qualify them to be widely used in different biological aspects [15]. The ability of silver nanoparticles to kill pathogens has been known for several years, making it widely used in various products. Silver nanoparticles can stop the G+ and G− bacteria population and exhibit a wide range of antibacterial action [16,17,18]. The discovery of antibacterial-activity nanoparticles that can inhibit different multidrug-resistant pathogens such as multidrug-resistant *P. aeruginosa* and ampicillin-resistant *E. coli* is significant in the medical field. Silver nanoparticles can also inhibit fungi growth including *A. fumigatus*, *Mucor*, *Candida tropicalis,* and *S. cerevisiae* [19]. Silver nanoparticles have antiviral activity; according to several studies, AgNPs prevented the replication and activity of HIV-1 by 98%, which was substantially higher than Au nanoparticles (6–20%) [20]. Additionally, it has an inhibitory effect against the hepatitis B virus [20] and the herpes simplex virus [21]. Nanosilver has excellent antibacterial activity, which is why it is a growing are of study and has been heavily industrialized. It can be found in a vast range of products on the consumer market. Different effects can be caused by physically or chemically synthesized AgNPs including DNA damage, developmental abnormalities, variation in the gene expression, and disturbance of the metabolism depending on concentration. The toxicity of AgNPs varies significantly depending on the shape, size, exposure pathway, and capping agents [22].

Many studies have produced and applied AgNPs in different ways. Song et al. [23] fabricated Ag nanoparticles/attapulgite nanocomposites using an olive leaf extract with antioxidant and antibacterial activity, and Sun et al. [24] fabricated silver nanoparticles combined with graphene oxide from table olive and used as a pollutant adsorbent.

No studies have shed light on using olive leaf waste in fabricating AgNPs, and then applied in medical uses. Therefore, in this study, we valorized the accumulated olive leaf waste that poses a danger to the health and environment by fabricating safe silver nanoparticles (OLAgNPs), then used them as an antioxidant, anticancer, and antimicrobial agent. Six advanced devices characterize the produced AgNPs. HPLC evaluated the phenolic compound profile in OLAgNPs and OLWE as well as the antioxidant activity, antitumor activity against the MCF-7, HeLa, and HT-29 carcinoma cell lines, and antimicrobial activity against MDR bacteria and fungi was evaluated.

## 2. Materials and Methods

### 2.1. Materials

Olive leaf wastes were obtained from a private olive farm in Ismaillia, Egypt (30.482811, 32.250064). The wastes were washed. All chemicals in this study were of analytical grade and obtained from Sigma (Cairo, Egypt). The biodiagnostic company (Dokki, Giza, Egypt) provided the cancer cell lines. The microbial strains, *Listeria monocytogenes* (LM), *Bacillus cereus* (BC), *Staphylococcus aureus* (SA), *Yersinia enterocolitica* (YE), *Campylobacter jejuni* (CJ), and *Escherichia coli* (EC) and six pathogenic fungi (i.e., *Candida glabrata* (CG), *Candida rugosa* (CR), *Candida albicans* (CA), *Penisillum crustosum* (PC), *Aspergillus niger* (AN), and *Aspergillus flavus* (AF)) were obtained from the Faculty of Agriculture, Ain-Shams University, Egypt.

### 2.2. Extracting Phenolic Compounds from OLWE

First, the olives were washed to remove physical impurities such as leaves, pieces of wood as well as any pesticides. Afterward, the olives were ground and mixed into a paste. Although many extracting systems are available, two methods are generally employed: traditional pressing and modern centrifuging. All wastes were collected and called OLWE. The OLWE was then kept in a 50 °C oven for two days, crushed in a blender, and then crushed in a blender, sieved through a 20-mesh screen. A total of 10 g of crushed leaves were stirred in distilled H_2_O (150 mL) for four hours at 40 °C. The supernatant was filtrated and then concentrated by a rotary evaporator (Figure 1). The samples were freeze-dried, and then their weight was recorded to calculate the total extract yield (%) from Equation (1).
(1)$$\mathrm{Total}\;\mathrm{extract}\;\mathrm{yield}\;\left(\%\right)=\frac{\mathrm{weight}\;\mathrm{of}\;\mathrm{dried}\;\mathrm{extract}}{100\;\mathrm{grams}\;\mathrm{of}\;\mathrm{OLWE}}\times 100\;$$

### 2.3. Green Synthesis of AgNPs

A total of 100 mg of OLWE-rich phenolic extracts were blended and stirred in 100 mL of 1 mM AgNO_3_ solution for 12 h at room temperature. The centrifugation of the OLAgNP suspension was conducted at 13,000 rpm, 15 min using Sigma 3–k, Germany centrifuge (Sigma, Offenbach, Germany). The supernatant was removed, the precipitate containing AgNPs was washed many times with ddH2O, centrifuged, and pure pellets were obtained (Figure 1).

### 2.4. Characterization of Green OLAgNPs

The OLAgNPs were characterized using six advanced instruments. The optical property of AgNPs was determined by UV–VIS. Spectroscopy examination utilized a Laxco^TM^ dual-beam spectrophotometer (Los Angeles, CA, USA) between 200 and 700 nm. A transmission electron microscope (JEOL, 1010, Tokyo, Japan) was used to determine the size and form of AgNPs. The active groups in the OLWE and OLAgNPs were detected in the IR spectrum (4000–400 cm^−1^) using FTIR (Bruker Tensor 37, Kaller, Bremen, Germany). The dried portion from the samples was mixed with KBR before measuring. A Nano “Z2 Malvern (Malvern Hills, Worcester, UK) was utilized for the DLS analysis. The net surface charge of the OLAgNPs was measured by the zeta potential (Malvern Hills, Worcester, UK), ensuring the stability of the generated nanoparticles while the average size was determined by a zeta sizer (Malvern Hills, Worcester, UK).

### 2.5. Phenolic Content

#### 2.5.1. Evaluation of Total Phenolic Compounds and Total Flavonoids

According to Saad et al. [25], the total polyphenol (TP) and total flavonoid (TF) content of the OLWE and OLAgNPs were estimated using a microtiter plate reader (BioTek Elx808, Santa Clara, CA, USA). Concerning the TPs, 50 µL of the OLWE or OLAgNPs were loaded in the plate well, then 100 µL of Na_2_CO_3_ and Folin reagent was added, and the developed blue color was measured at 750 nm; the TPs were calculated as mg GAE/g. Regarding TFs, 50 µL of the OLWE or OLAgNPs were loaded in the plate well, then 100 µL of ethanolic AlCl_3_ was added, and the yellow color was read at 430 nm and expressed as mg QE/g.

#### 2.5.2. HPLC Profile for Phenolic Compounds in OLWE or OLAgNPs

The HPLC (Shimadzu20A, Tokyo, Japan) was applied to assess the phenolic compounds in the OLWE and OLAgNPs. The mobile phase constitutes a 0.01% acetic acid solution in aqua (A) and acetonitrile (B). The stationary phase was the C18 separation column (Gemini, 4.6 × 150 mm × 5 µm).The flow rate was adjusted to 2 mL/min. For the first 5 min, the isocratic elution was 95% A and 5% B, followed by a linear gradient to 50% A and 50% B within 5–55 min, then 50% A and 50% B within 55–65 min, and 95% A and 5% B at 65–67 min; post-time was 6 min before the next injection. The HPLC parts, autosampler, pumps, column, oven, and diode array system were adjusted and monitored. The collected results were analyzed using Class VP software (Shimadzu 5.0, Japan). At 280 nm, the phenolics were estimated, and at 370 nm, the flavonoids [26,27].

### 2.6. The Biological Activity of OLWE and OLAgNPs

#### 2.6.1. Antioxidant

The OLAgNPs and OLWE were tested for their ability to scavenge DPPH at a concentration of 50 µg/mL following Saad et al. [25]. First, 100 µL of AgNPs and OLWE were homogenized in 100 µL of DPPH for 10 min; the mix was placed in wells of a microtiter plate and kept in the dark for 30 min; the ready plate was read at 517 nm using a microtiter plate reader (BioTek Elx808, Santa Clara, CA, USA) and the equation was applied (Equation (2)).
(2)$$\%\;\mathrm{Antioxidant}\;\mathrm{activity}=\frac{\mathrm{Control}\;\mathrm{absorbance}-\mathrm{sample}\;\mathrm{absorbance}}{\mathrm{Control}\;\mathrm{absorbance}}\times 100\;$$

#### 2.6.2. Antimicrobial

The antimicrobial activity of the OLAgNPs and OLWE was performed against various bacterial strains. The activity of the OLAgNPs and OLWE was also tested against fungi. The bacterial and *Candida* strains were selected depending on the microbiological testing of patients infected with microorganisms. The bacteria and *Candida* strains were cultured overnight at 37 °C in a shaking incubator with Muller Hinton broth (MHB) to a concentration of 1 × 10^8^ CFU/mL. The disc diffusion method was used to conduct the antibacterial activity [28]. The plates were inculcated using the spread plate technique with 100 µL of the active bacterial and *Candida* isolates. Previously saturated paper discs (6 mm) with OLAgNPs (5, 25, 50, and 100 µg/mL) and OLWE (100 μg/mL) were placed on the surface of the plates. The plates were incubated for 24 h at 37 °C. A ruler measured the inhibition zones surrounding the discs (mm). An antifungal antibiotic and levofloxacin as a positive control.

#### 2.6.3. Cytotoxicity Effects

The sulforhodamine B (SRB) assay was used to evaluate the viability of the MCF-7, HeLa, and HT-29 cell lines. Cancer cells (5 × 10^3^ cells) were suspended in 100 µL of the control medium and cultured for 24 h in 96-well plates. Under the same circumstances, another 100 µL was added to the medium supplemented with various doses of OLAgNPs (5, 25, 50, and 100 µg/mL) and OLWE (100 µg/mL). The samples were treated with 150 µL of 10% TCA for three days to ensure fixation, then incubated for one hour at 4 °C. Multiple times, the cells were rinsed with dH_2_O. A total of 70 µL of SRB solution (0.4% *w/v*) was then added and incubated for 10 min in the dark. The plates were cleaned with acetic acid (1%) three times before being air-dried for an entire night. The protein-bound SRB dye was dissolved with 150 µL of 10 mM TRIS-HCl; then, the absorbance of the developed color was read at 540 nm using a microtiter plate reader (BioTek Elx808, USA). The LC_50_ was established as the medication concentration that caused a 50% decrease in absorbance relative to the control cells [29].

### 2.7. Statistical Analyses

The data analysis was performed using the Biostat 2009 software (version 5.8.4.3, 2010). ANOVA was used to examine the data from the biological studies, and the LSD test was used to analyze the means at *p* ≤ 0.05 [30].

## 3. Results

### 3.1. Green Synthesis of AgNPs from Phenolic Extracts

The silver nitrate solution turned the light green color of OLWE into dark brown after 24 h. Adding Ag+ to OLWE changed the color from green to dark brown, indicating that metal ions had been reduced and silver nanoparticles had formed. Adding AgNO_3_ to OLWE (1:100 *v/v*) provided the highest yield of OLAgNPs. The conversion of AgNPs without precipitation revealed that the particle sizes of the synthetic nanoparticles were incredibly tiny.

### 3.2. OLAgNP Characterization

The UV–Vis spectrum revealed that OLWE produced varying quantities of OLAgNPs, confirming the synthesis and durability of the AgNPs. Using a Laxco^TM^ dual-beam spectrophotometer, the absorption spectra of nanoparticles were recorded after a day in the 200–700 nm region. The UV and TEM results in Figure 1 revealed that the OLWE produced various concentrations of AgNPs. The highest UV absorbance was at 400 nm with 0.9 a.u, recorded for OLAgNPs (Figure 1A). The TEM image in Figure 1B configures the morphological features of OLAgNPs. The average size of oval to spherical OLAgNPs (Appendix A) ranged from 20 to 45 nm. The results indicate that OLAgNPs were crystalline (Appendix A) and detected by EDS, where an obvious peak (30θ refers to Ag). In Figure 1C,D, the DLS results of the OLAgNPs revealed a single peak. The exact size of OLAgNPs was 28 nm, with a negative charge of −21 eV. The nanoparticles’ negative surface charge contributed to their stability. The FTIR spectra of OLAgNPs and OLWE were from 4000 to 400 cm^−1^, as shown in Figure 1E,F. Only eight bands were detected in the OLWE spectra, which increased to thirteen bands in the FTIR spectra of OLAgNPs. The OH groups were found at 3450 cm^−1^, the NH group at 3300 cm^−1^, 1750 cm^−1^ referred to aldehyde, C=N was found at 1570 cm^−1^, CH_3_, COH, and C–O–C were found between 1054 and 1360 cm^−1^. The spectra indicated that the examined AgNPs had reducing polyphenols in the extract on their surface. Consequently, the FTIR analyses indicated that the secondary metabolites of the plant leaf extract successfully encapsulated and stabilized the green-produced silver nanoparticles.

### 3.3. Phenolic Compounds in OLAgNPs and OLWE

The total phenolic and flavonoid content in the olive leaf waste extracts and silver nanoparticles are presented in Figure 2. The phenolic compounds in the AgNPs increased in a concentration-dependent manner. The TPs of OLAgNPs (100 µg/mL) increased by 57% over OLWE, while the TFs increased by 33%.

The phenolic compound profile in the OLWE and OLAgNPs are presented in Table 1. The FTIR profile indicates the presence of phenolic compounds in a nanoparticle suspension (Figure 1). Gallic acid, chlorogenic acid, rutin, naringenin, catechin, and propyl gallate were the main compounds in the HPLC profile of the OLAgNPs and OLWE. The phenolic compounds significantly increased in AgNPs over OLWE, with a relative increase ranging between 5.71% in innaringenin and 66% in vanillin over the OLWE. The phenolic compounds were divided into three groups based on the percentage of increase, Ι (>50%), ΙΙ (30–50%), and ΙΙΙ (<30%) (Table 1).

### 3.4. Biological Activities of AgNPs and OLWE

#### 3.4.1. Antioxidant Activity of AgNPs and OLWE

The antioxidant activity of AgNPs and OLWE is presented in Figure 3. The findings revealed that OLWE scavenged 91% of DPPH radicals, achieving no significant difference to ascorbic acid, which scavenged 92%, and AgN100, which scavenged 93% of DPPH˙. Because of the bioactive molecules on the surface of the AgNPs, the biosynthesized silver nanoparticles exhibited a superior antioxidant activity compared to the ascorbic acid.

#### 3.4.2. Cytotoxicity Effect of AgNPs and OLWE

The SRB assay was performed to determine the cytotoxicity of OLAgNPs and OLWE against three cancer cell lines: MCF 7 (breast cancer cell line), HT-29 (colon cancer cell line), and HeLa. The SRB test depends on the property of SRB, which binds to proteins in stoichiometry under mildly acidic conditions and can then be eliminated under basic conditions; hence, the quantity of bound dye may be used as a surrogate for cell mass, which can be extrapolated to quantify cell proliferation. The concentrations of AgNPs (5, 25, 50, and 100 µg/mL) and OLWE (100 µg/mL) were tested for their effects on the vitality and proliferation of the cancer cell lines to determine the metabolic reduction potency of the cancer cell lines (Figure 4). The microscopic image shows that the OLAgNPs had a high cytotoxic effect against the Hela, MCF-7, and HT-29 cell lines, inhibiting the cancer cell viability by 79–82%. The OLAgNP toxicity exceeded that of DOX and OLWE by 3–22%. Both the OLWE and AgNPs were discovered to have a cytotoxic effect on the cancer cell lines in a concentration-dependent manner with an IC_50_ of 5 µg/mL for OLAgNPs, 10 µg/mL for DOX, and 30 µg/mL for OLWE. The effect of AgNPs on the viability of the treated cell lines is presented in Figure 5, and clarified that a considerable decrease in cell viability was seen when the AgNP concentrations increased. Furthermore, with the lowest concentration of AgNPs, the maximum proportion of cell viability was observed (Figure 5). The increased efficacy of AgNPs generated using olive leaf extract against cancerous cells may be attributable to their smaller size, which promotes cell entrance. The phytochemicals of OLE improved the biocompatibility and cellular accessibility of AgNPs.

#### 3.4.3. Antimicrobial Activity of AgNPs and OLWE

AgNPs and OLWE were tested for antimicrobial activity against six studied MDR bacteria, i.e., *L. monocytogenes* (LM), *B. cereus* (BC), *S. aureus* (SA), *Y. enterocolitica* (YE), *C. jejuni* (CJ), and *E. coli* (EC) and six pathogenic fungi (i.e., *C. glabrata* (CG), *C. rugosa* (CR), *C. albicans* (CA), *P. crustosum* (PC), *A. niger* (AN), and *A. flavus* (AF)) using the disc diffusion assay. The inhibition zone diameter (mm) was measured. The results revealed the dose-dependent activity of AgNPs against pathogens (Table 2). The results in Table 2 demonstrated that bacteria exposed to low concentrations of OLWE (5, 25, and 50 µg/mL) did not exhibit any inhibitory zones in the agar well diffusion assay; however, the inhibition zones occurred at higher concentrations (100 µg/mL). At the same time, the AgNPs showed inhibition zones against different bacterial species exposed to low concentrations and, by extension, high concentrations. Moreover, the AgNPs and OLWE had similar inhibitory effects on pathogenic fungi. Additionally, the outcomes demonstrated that AgNPs at a higher concentration (100 µg/mL) exhibited greater inhibitory zones than the positive controls (standard antibiotic). The inhibition zone diameters against the MDR bacteria ranged from 25 to 37 nm compared to 16–26 in OLWE and 22–35 in antibiotics, indicating the higher potency of AgNPs against MDRB than the standard antibiotic. *L. monocytogenes* and *C. jejuni* were the most resistant bacteria to AgNPs, OLWE, and antibiotics. Similarly, our OLAgNPs had considerable antifungal activity in the range of 26–35 mm compared to 25–32 mm in fungal antibiotics. *C. glabrata* and *C. rugosa* were the most resistant fungi to AgNPs, OLWE, and antibiotics. Similar results demonstrated that olive oil-fabricated silver nanoparticles had the greatest antibacterial activity against K. pneumoniae with an inhibition zone diameter (IZD) of 39 mm, followed by *P. aeruginosa* with an IZD of 30 mm. *S. aureus, M. luteus, C. albican, S. flexneri,* and *P. mirabilis* were susceptible to olive oil (NPs) between 26 mm and 29 mm.

Figure 6 shows that OLAgNPs at a concentration of 2.5–10 µg/mL inhibited the growth of MDRB and pathogenic fungi, which killed them at a concentration of 10–20 µg/mL compared to the OLWE and antibiotics with a relative decrease of 50–65%.

## 4. Discussion

Olive leaves are a rich source of many biologically secondary metabolites that can be employed as reducing and capping agents to manufacture biogenic nanoparticles [31]. For manufacturing OLAgNPs, we valorized the olive leaf waste extract as a stabilizing and reducing agent. Sellami et al. [32] used the olive leaf extract to develop green silver nanoparticles; Omar et al. [33] also used olive pomace extracts as promising candidates for fabricating AgNPs, and by managing the elements affecting the suggested technique, produced a greater quantity of AgNPs with smaller particles, wherein the current study, no precipitations appeared in the OLAgNP solutions. Phenolic compounds in OLWE may be responsible for the reduction of silver ions to metallic form, which is a potential mechanism for creating OLAgNPs. Other secondary metabolites in the leaf extract including terpenoids, saponins, and alkaloids may quickly stabilize the produced Ag0 by capping and transforming it into silver nanoparticles [34].

Concerning the characterization, our AgNPs were similar to the fabricated AgNPs with olive leaves, and showed absorbance at 410 nm for AgNPs and 310 nm for polyphenols [32,35]. The concentration of silver nitrate, temperature conditions, and phenolic content in the extracts used to reduce silver ions were the key factors influencing the size difference [35]. In this regard, Felimban et al. [36] biosynthesized AgNPs that were spherical, with small sizes ranging from 13 to 21 nm and were highly stable at −23 and −24 mV.

The active groups that are responsible for nanoparticle transformation detected by FTIR, as reported by Sellami et al. [32] for olive leaf extract, were exhibited different IR peaks, the phenolic OH was detected at 3400 cm^−1^, C–H at 2920–2880 cm^−1^, carbonyl at 1706 cm^−1^ and 1650 cm^−1^ (aromatic C=C), and other active groups [31], indicating the presence of numerous phyto-molecules, also found in produced AgNPs but with small variations in the wavenumbers and intensities that agreed with our results.

The phenolic content in nanoparticles increased in the extract due to the quinoid chemical generated by OH oxidation in the phenolics on the surface of the nanoparticles; the amount of polyphenols in nanoparticles has increased, boosting the phenolic content and assuring stability for nanoparticle suspension [37]. In this regard, Saad et al. [25] reported that silver nanoparticles produced from pomegranate and watermelon peel extract were rich in phenolic compounds, with the phenolic content in the AgNPs increasing by 10–75% compared to the extract. Principal phenolic components were quercetin, gallic acid, catechin, cyanidin-3-O-glu, punicalagin, and ellagic acid.

AgNPs may thus be used as antioxidants in the future since they are very effective, non-toxic, environmentally friendly, and affordable [38]. Our results agreed with Sreeleka et al. [39]: green nanoparticles exhibited 91% scavenging activity at a concentration of 5 µg/mL, whereas chemically produced nanoparticles exhibited only 79% at the same dosage. Due to a bioactive capping agent on the surface of the nanoparticles generated by *M. frondosa* leaf extract, their antioxidant activity was superior to that of chemically fabricated nanoparticles, also,

Regarding anticancer activity, our AgNPs were higher than the study by [36], the IC_50_ values for T47D cells in the presence of AgNPs were 116 g/mL, 176 g/mL in the presence of OLE, and 84 g/mL in the presence of AgNPs-OLE. The anticancer mechanism of AgNPs may depend on disrupting the mitochondrial respiration chain, inducing the generation of reactive oxygen species that damage DNA [40].

The silver nanoparticles synthesized with sunflower oil exhibited the greatest antibacterial activity against *K. pneumoniae*, with an inhibition zone of 38.90 mm, followed by *P. aeruginosa, S. aureus, M. luteus, C. albican,* and *P. mirabilis*, each with an inhibition zone of between 25.90 and 38.90 mm (28 mm to 26 mm). The sensitivity of *S. flexneri* to sunflower oil (NPs) with an inhibitory zone was the lowest (26 mm) [41].

Nanoparticles can ionize and liberate Ag+ into the reaction media [42]. The formation of holes in the phospholipid layer of the bacterial cell membrane may be an antibacterial mechanism for AgNPs; consequently, their instability and membrane permeability increase [43]. Silver also builds up in the cell wall or membrane, creating holes. The functional groups in the active core of enzymes and cell membrane proteins allow silver to attach to the cell wall, so the content of ROS produced by bacteria increases [44], causing cell protein denaturation, the unregulated transit of ions, and the leakage of protons and metabolites from bacterial cells, or all three [45].

## 5. Conclusions

Accumulating agricultural wastes cause environmental and health problems; however, they are rich in polyphenols, so in this study, we valorized OLW to produce OLWE that can reduce silver nitrate into green silver nanoparticles. Due to the presence of bioactive groups in OLWE, they surrounded the surface of the AgNPs; the biosynthesized silver nanoparticles exhibited superior antioxidant, antimicrobial, and anticancer activities. Green and safe AgNPs may thus be used as antioxidants in the future since they are very effective, non-toxic, environmentally friendly, and affordable; they also can be used as an alternative antibiotic against MDR microorganisms that may be included in cancer medical formulations.

## Data Availability

Not applicable.

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
