# Peer review of "Green Biogenic of Silver Nanoparticles Using Polyphenolic Extract of Olive Leaf Wastes with Focus on Their Anticancer and Antimicrobial Activities"

_plants, 2023, doi:10.3390/plants12061410_

Round 1

Reviewer 1 Report

The MS has a title

“Fabricating green silver nanoparticles by polyphenolic olive leaves wastes' extract and evaluating their anticancer and antimicrobial activities”

The green production of silver nanoparticles is a common publication which already reported by several researchers like:

- Alhajri, H.M.; Aloqaili, S.S.; Alterary, S.S.; Alqathama, A.; Abdalla, A.N.; Alzhrani, R.M.; Alotaibi, B.S.; Alsaab, H.O. Olive Leaf Extracts for a Green Synthesis of Silver-Functionalized Multi-Walled Carbon Nanotubes. J. Funct. Biomater. 2022, 13, 224. https://doi.org/10.3390/jfb13040224

- Sun W, Yaoliang Hong, Tian Li, Huaqiang Chu, Junxia Liu, Li Feng, Mehidi Baghayeri, (2023). Biogenic synthesis of reduced graphene oxide decorated with silver nanoparticles (rGO/Ag NPs) using table olive (olea europaea) for efficient and rapid catalytic reduction of organic pollutants. Chemosphere, 310, 136759. https://doi.org/10.1016/j.chemosphere.2022.136759.

This was started from several years ago like the following ref., and many other plants were used in the biosynthesis

Mostafa M. H. Khalil Eman H. IsmailDoaa Mohamed (2014). Green synthesis of silver nanoparticles using olive leaf extract and its antibacterial activity, Arabian Journal of Chemistry, https://doi.org/10.1016/j.arabjc.2013.04.007

The MS has many comments:

1- Starting from the title, may some changes in the title are needed, not so good to write many words in plural (with s), in the title like “leaves wastes”

The suggested one is:

“Green Biogenic of Silver Nanoparticles using Polyphenolic Extract of Olive Leaf Wastes with Focus on their Anticancer and Antimicrobial activities”

2- Keywords, please do not repeat any word already mentioned in the title?

3- Introduction, this is not acceptable section in the MS, simply please collect all paragraphs that are talking on one theme in one paragraph, ranging from 3-5 paragraphs, like the first on Silver nanoparticles (AgNPs), the second paragraph will be on olive leaves waste, an d the third may be on the anti-microbial activities, or the green o synthesize the nanoparticles!!!

Not two or three sentences giving one paragraph (lines 49 -52)?

This section must include very recent refs. like published on 2023 and 2022, please

4- Materials and methods, where and when this study was carried out? Which Lab?? the source of olive leaf wastes?

the procedure steps for preparing this nanoparticle should presented in one figure please!

5- please, any instrument must be mentioned in the MS, model, the city, the country of manufacture, (lines 118-127)

“The OLAgNPs were characterized using six advanced instruments…..”

Transmission Electron Microscope (Japan) was used to determine the size and form of AgNPs, (where the figures or photos?) or SEM and EDS Measurements, where, please!

6- the same sore any chemicals, sources, purity (if any), the country of manufacture

7- any method must be followed by the ref., please like measurement of antioxidants (where the ref.?), the equations, where their sources?

8- line 183, (2.6. Data analysis, change into Statistical analyses), please

9- It is preferable to separate the “3. Results and discussion” into two sections, please to present only results in its section and discussion in the second section, please

10- In tables 1 and 2, please add the ± SD, and which number (N)?

11- Fig 6 need to add Duncan (letter) above each column, please

Author Response

Response to Reviewer (1) Comments

Manuscript ID: plants-2250001

Manuscript Title: Fabricating green silver nanoparticles by polyphenolic olive leaves wastes' extract and evaluating their anticancer and antimicrobial activities

We are very glad that Reviewer (1) highly evaluated our manuscript, and provided constructive comments and valuable suggestions that have helped us further improvement of the quality of our manuscript. All the changes made in response to the Reviewer (1) comments were corrected as track changes in the revised manuscript. We have addressed all of your queries and improved our manuscript following your suggestions as you can see in our point-to-point responses to your comments below.

Comments and Suggestions for Authors

The MS has a title

“Fabricating green silver nanoparticles by polyphenolic olive leaves wastes' extract and evaluating their anticancer and antimicrobial activities”

 The green production of silver nanoparticles is a common publication which already reported by several researchers like:

- Alhajri, H.M.; Aloqaili, S.S.; Alterary, S.S.; Alqathama, A.; Abdalla, A.N.; Alzhrani, R.M.; Alotaibi, B.S.; Alsaab, H.O. Olive Leaf Extracts for a Green Synthesis of Silver-Functionalized Multi-Walled Carbon Nanotubes. J. Funct. Biomater. 2022, 13, 224. https://doi.org/10.3390/jfb13040224

- Sun W, Yaoliang Hong, Tian Li, Huaqiang Chu, Junxia Liu, Li Feng, MehidiBaghayeri, (2023). Biogenic synthesis of reduced graphene oxide decorated with silver nanoparticles (rGO/Ag NPs) using table olive (olea europaea) for efficient and rapid catalytic reduction of organic pollutants. Chemosphere, 310, 136759. https://doi.org/10.1016/j.chemosphere.2022.136759.

This was started from several years ago like the following ref., and many other plants were used in the biosynthesis

Mostafa M. H. Khalil Eman H. IsmailDoaa Mohamed (2014). Green synthesis of silver nanoparticles using olive leaf extract and its antibacterial activity, Arabian Journal of Chemistry, https://doi.org/10.1016/j.arabjc.2013.04.007

 Response: Thanks for the reviewer. We appreciate and consider all comments. Considering, linguistically mistakes, the manuscript was extensively revised by an expert and highlight with red color

The novelty of our work was generated from valorizing olive leaves waste to produce OLAgNPs, not using the leaf extract. Valorizing waste is a worldwide trend for a safe environment and human health

The MS has many comments:

1- Starting from the title, may some changes in the title are needed, not so good to write many words in plural (with s), in the title like “leaves wastes”

The suggested one is:

“Green Biogenic of Silver Nanoparticles using Polyphenolic Extract of Olive Leaf Wastes with Focus on their Anticancer and Antimicrobial activities”

 Response: Thanks for the reviewer; we consider this change

2- Keywords, please do not repeat any word already mentioned in the title?

Response: Done as requested

3- Introduction, this is not acceptable section in the MS, simply please collect all paragraphs that are talking on one theme in one paragraph, ranging from 3-5 paragraphs, like the first on Silver nanoparticles (AgNPs), the second paragraph will be on olive leaves waste, and the third may be on the anti-microbial activities, or the green o synthesize the nanoparticles!!!

Not two or three sentences giving one paragraph (lines 49 -52)?

This section must include very recent refs. like published on 2023 and 2022, please

Response: The introduction was rearranged and updated accordingly

4- Materials and methods, where and when this study was carried out? Which Lab?? the source of olive leaf wastes?

Response: Done as requested

the procedure steps for preparing this nanoparticle should presented in one figure please!

Response: Done as requested

5- please, any instrument must be mentioned in the MS, model, the city, the country of manufacture, (lines 118-127)

Response: Done as requested

“The OLAgNPs were characterized using six advanced instruments…..”

Transmission Electron Microscope (Japan) was used to determine the size and form of AgNPs, (where the figures or photos?) or SEM and EDS Measurements, where, please!

Response: TEM image exists in Figure 1B. We also add supplementary characterization with SEM and EDS

6- the same sore any chemicals, sources, purity (if any), the country of manufacture

Response: Done as requested

7- any method must be followed by the ref., please like measurement of antioxidants (where the ref.?), the equations, where their sources?

Response: Done as requested

8- line 183, (2.6. Data analysis, change into Statistical analyses), please

Response: Done as requested

9- It is preferable to separate the “3. Results and discussion” into two sections, please to present only results in its section and discussion in the second section, please

Response: Done as requested

10- In tables 1 and 2, please add the ± SD, and which number (N)?

Response: Done as requested

11- Fig 6 need to add Duncan (letter) above each column, please

Response: Done as requested

Reviewer 2 Report

Row 112 "100 mg of OLWE phenolic extracts" - this is dried water extract is not phenolic. If you considered his like phenolic extract please provide methods used to separate just phenolic compounds. Your extract could a rich phenolic extract. 

Row 191 "However, to manufacture OLAgNPs, we valorized the olive leaf waste extract" - Olive leaf waste extract is represented by the waste from other technological process? Present some details about this olive leaf waste.

Row 327 "higher concentration (100 g/mL) exhibited" - probably is 100 micrograms/mL

Row 344-345 - the format is different by the entire text

Author Response

Response to Reviewer (2) Comments

Manuscript ID: plants-2250001

Manuscript Title: Fabricating green silver nanoparticles by polyphenolic olive leaves wastes' extract and evaluating their anticancer and antimicrobial activities

We are very glad that Reviewer (2) highly evaluated our manuscript, and provided constructive comments and valuable suggestions that have helped us further improvement of the quality of our manuscript. All the changes made in response to the Reviewer (2) comments were corrected as track changes in the revised manuscript. We have addressed all of your queries and improved our manuscript following your suggestions as you can see in our point-to-point responses to your comments below.

Comments and Suggestions for Authors

Row 112 "100 mg of OLWE phenolic extracts" - this is dried water extract is not phenolic. If you considered his like phenolic extract please provide methods used to separate just phenolic compounds. Your extract could a rich phenolic extract.

 Response: Thanks for the reviewer. We adjusted ti OLWE rich phenolic extract

Row 191 "However, to manufacture OLAgNPs, we valorized the olive leaf waste extract" - Olive leaf waste extract is represented by the waste from other technological process? Present some details about this olive leaf waste.

Response: Done as requested, we cleared that in material and methods, additionally; added scheme 1 cleared the experimental design of our study

Row 327 "higher concentration (100 g/mL) exhibited" - probably is 100 micrograms/mL

Response: it was adjusted

Row 344-345 - the format is different by the entire text

Response: it was adjusted

Round 2

Reviewer 1 Report

Many thanks for improving the MS

All the best!